# Matrix Manifold Optimization for Gaussian Mixtures

**Reshad Hosseini**
School of ECE
College of Engineering
University of Tehran, Tehran, Iran
reshad.hosseini@ut.ac.ir

**Suvrit Sra**
Laboratory for Information and Decision Systems
Massachusetts Institute of Technology
Cambridge, MA.
suvrit@mit.edu

## Abstract

We take a new look at parameter estimation for Gaussian Mixture Model (GMMs). Specifically, we advance *Riemannian manifold optimization* (on the manifold of positive definite matrices) as a potential replacement for Expectation Maximization (EM), which has been the *de facto* standard for decades. An out-of-the-box invocation of Riemannian optimization, however, fails spectacularly: it obtains the same solution as EM, but vastly slower. Building on intuition from geometric convexity, we propose a simple reformulation that has remarkable consequences: it makes Riemannian optimization not only match EM (a nontrivial result on its own, given the poor record nonlinear programming has had against EM), but also outperform it in many settings. To bring our ideas to fruition, we develop a well-tuned Riemannian LBFGS method that proves superior to known competing methods (e.g., Riemannian conjugate gradient). We hope that our results encourage a wider consideration of manifold optimization in machine learning and statistics.

## 1 Introduction

Gaussian Mixture Models (GMMs) are a mainstay in a variety of areas, including machine learning and signal processing [4, 10, 16, 19, 21]. A quick literature search reveals that for estimating parameters of a GMM the Expectation Maximization (EM) algorithm [9] is still the *de facto* choice. Over the decades, other numerical approaches have also been considered [24], but methods such as conjugate gradients, quasi-Newton, Newton, have been noted to be usually inferior to EM [34].

The key difficulty of applying standard nonlinear programming methods to GMMs is the positive definiteness (PD) constraint on covariances. Although an open subset of Euclidean space, this constraint can be difficult to impose, especially in higher-dimensions. When approaching the boundary of the constraint set, convergence speed of iterative methods can also get adversely affected. A partial remedy is to remove the PD constraint by using Cholesky decompositions, e.g., as exploited in semidefinite programming [7]. It is believed [30] that in general, the nonconvexity of this decomposition adds more stationary points and possibly spurious local minima.[1] Another possibility is to formulate the PD constraint via a set of smooth convex inequalities [30] and apply interior-point methods. But such sophisticated methods can be extremely slower (on several statistical problems) than simpler EM-like iterations, especially for higher dimensions [27].

Since the key difficulty arises from the PD constraint, an appealing idea is to note that PD matrices form a Riemannian manifold [3, Ch.6] and to invoke *Riemannian manifold optimization* [1, 6]. Indeed, if we operate on the manifold[2], we implicitly satisfy the PD constraint, and may have a better chance at focusing on likelihood maximization. While attractive, this line of thinking also fails: an out-of-the-box invocation of manifold optimization is also vastly inferior to EM. Thus, we need to think harder before challenging the hegemony of EM. We outline a new approach below.

**Key idea.** Intuitively, the mismatch is in the geometry. For GMMs, the M-step of EM is a *Euclidean convex* optimization problem, whereas the GMM log-likelihood is not *manifold convex*[3] even for a single Gaussian. If we could reformulate the likelihood so that the single component maximization task (which is the analog of the M-step of EM for GMMs) becomes manifold convex, it might have a substantial empirical impact. This intuition supplies the missing link, and finally makes Riemannian manifold optimization not only match EM but often also greatly outperform it.

To summarize, the key contributions of our paper are the following:

– Introduction of Riemannian manifold optimization for GMM parameter estimation, for which we show how a reformulation based on geodesic convexity is crucial to empirical success.
– Development of a Riemannian LBFGS solver; here, our main contribution is the implementation of a powerful line-search procedure, which ensures convergence and makes LBFGS outperform both EM and manifold conjugate gradients. This solver may be of independent interest.

We provide substantive experimental evidence on both synthetic and real-data. We compare manifold optimization, EM, and unconstrained Euclidean optimization that reformulates the problem using Cholesky factorization of inverse covariance matrices. Our results shows that manifold optimization performs well across a wide range of parameter values and problem sizes. It is much less sensitive to overlapping data than EM, and displays much less variability in running times.

Our results are quite encouraging, and we believe that manifold optimization could open new algorithmic avenues for mixture models, and perhaps other statistical estimation problems.

**Note.** To aid reproducibility of our results, MATLAB implementations of our methods are available as a part of the MIXEST toolbox developed by our group [12]. The manifold CG method that we use is directly based on the excellent toolkit MANOPT [6].

**Related work.** Summarizing published work on EM is clearly impossible. So, let us briefly mention a few lines of related work. Xu and Jordan [34] examine several aspects of EM for GMMs and counter the claims of Redner and Walker [24], who claimed EM to be inferior to generic second-order nonlinear programming techniques. However, it is now well-known (e.g., [34]) that EM can attain good likelihood values rapidly, and scales to much larger problems than amenable to second-order methods. Local convergence analysis of EM is available in [34], with more refined results in [18], who show that when data have low overlap, EM can converge locally superlinearly. Our paper develops Riemannian LBFGS, which can also achieve local superlinear convergence.

For GMMs some innovative gradient-based methods have also been suggested [22, 26], where the PD constraint is handled via a Cholesky decomposition of covariance matrices. However, these works report results only for low-dimensional problems and (near) spherical covariances.

The idea of using manifold optimization for GMMs is new, though manifold optimization by itself is a well-developed subject. A classic reference is [29]; a more recent work is [1]; and even a MATLAB toolbox exists [6]. In machine learning, manifold optimization has witnessed increasing interest[4], e.g., for low-rank optimization [15, 31], or optimization based on geodesic convexity [27, 33].

## 2  Background and problem setup

The key object in this paper is the *Gaussian Mixture Model (GMM)*, whose probability density is

$$p(\boldsymbol{x}) := \sum_{j=1}^{K} \alpha_j p_{\mathcal{N}}(\boldsymbol{x}; \boldsymbol{\mu}_j, \boldsymbol{\Sigma}_j), \qquad \boldsymbol{x} \in \mathbb{R}^d,$$

and where $p_{\mathcal{N}}$ is a (multivariate) Gaussian with mean $\boldsymbol{\mu} \in \mathbb{R}^d$ and covariance $\boldsymbol{\Sigma} \succ 0$. That is,

$$p_{\mathcal{N}}(\boldsymbol{x}; \boldsymbol{\mu}, \boldsymbol{\Sigma}) := \det(\boldsymbol{\Sigma})^{-1/2} (2\pi)^{-d/2} \exp\big(-\tfrac{1}{2}(\boldsymbol{x} - \boldsymbol{\mu})^T \boldsymbol{\Sigma}^{-1}(\boldsymbol{x} - \boldsymbol{\mu})\big).$$

Given i.i.d. samples $\{\boldsymbol{x}_1, \ldots, \boldsymbol{x}_n\}$, we wish to estimate $\{\hat{\boldsymbol{\mu}}_j \in \mathbb{R}^d, \hat{\boldsymbol{\Sigma}}_j \succ 0\}_{j=1}^{K}$ and weights $\hat{\boldsymbol{\alpha}} \in \Delta_K$, the $K$-dimensional probability simplex. This leads to the *GMM optimization* problem

$$\max_{\boldsymbol{\alpha} \in \Delta_K, \{\boldsymbol{\mu}_j, \boldsymbol{\Sigma}_j \succ 0\}_{j=1}^{K}} \quad \sum_{i=1}^{n} \log\Big(\sum_{j=1}^{K} \alpha_j p_{\mathcal{N}}(\boldsymbol{x}_i; \boldsymbol{\mu}_j, \boldsymbol{\Sigma}_j)\Big). \tag{2.1}$$

Solving Problem (2.1) can in general require exponential time [20].[5] However, our focus is more pragmatic: similar to EM, we also seek to efficiently compute local solutions. Our methods are set in the framework of manifold optimization [1, 29]; so let us now recall some material on manifolds.

## 2.1 Manifolds and geodesic convexity

A smooth manifold is a non-Euclidean space that locally resembles Euclidean space [17]. For optimization, it is more convenient to consider Riemannian manifolds (smooth manifolds equipped with an inner product on the tangent space at each point). These manifolds possess structure that allows one to extend the usual nonlinear optimization algorithms [1, 29] to them.

Algorithms on manifolds often rely on *geodesics*, i.e., curves that join points along shortest paths. Geodesics help generalize Euclidean convexity to *geodesic convexity*. In particular, say $\mathcal{M}$ is a Riemannian manifold, and $x, y \in \mathcal{M}$; also let $\gamma$ be a geodesic joining $x$ to $y$, such that

$$\gamma_{xy} : [0, 1] \to \mathcal{M}, \quad \gamma_{xy}(0) = x, \ \gamma_{xy}(1) = y.$$

Then, a set $\mathcal{A} \subseteq \mathcal{M}$ is *geodesically convex* if for all $x, y \in \mathcal{A}$ there is a geodesic $\gamma_{xy}$ contained within $\mathcal{A}$. Further, a function $f : \mathcal{A} \to \mathbb{R}$ is geodesically convex if for all $x, y \in \mathcal{A}$, the composition $f \circ \gamma_{xy} : [0, 1] \to \mathbb{R}$ is convex in the usual sense.

The manifold of interest to us is $\mathbb{P}^d$, the manifold of $d \times d$ symmetric positive definite matrices. At any point $\boldsymbol{\Sigma} \in \mathbb{P}^d$, the tangent space is isomorphic to set of symmetric matrices; and the Riemannian metric at $\boldsymbol{\Sigma}$ is given by $\text{tr}(\boldsymbol{\Sigma}^{-1} d\boldsymbol{\Sigma} \boldsymbol{\Sigma}^{-1} d\boldsymbol{\Sigma})$. This metric induces the geodesic [3, Ch. 6]

$$\gamma_{\boldsymbol{\Sigma}_1, \boldsymbol{\Sigma}_2}(t) := \boldsymbol{\Sigma}_1^{1/2}(\boldsymbol{\Sigma}_1^{-1/2}\boldsymbol{\Sigma}_2\boldsymbol{\Sigma}_1^{-1/2})^t\boldsymbol{\Sigma}_1^{1/2}, \quad 0 \le t \le 1.$$

Thus, a function $f : \mathbb{P}^d \to \mathbb{R}$ if geodesically convex on a set $\mathcal{A}$ if it satisfies

$$f(\gamma_{\boldsymbol{\Sigma}_1, \boldsymbol{\Sigma}_2}(t)) \le (1 - t)f(\boldsymbol{\Sigma}_1) + tf(\boldsymbol{\Sigma}_2), \quad t \in [0, 1], \ \boldsymbol{\Sigma}_1, \boldsymbol{\Sigma}_2 \in \mathcal{A}.$$

Such functions can be nonconvex in the Euclidean sense, but are globally optimizable due to geodesic convexity. This property has been important in some matrix theoretic applications [3, 28], and has gained more extensive coverage in several recent works [25, 27, 33].

We emphasize that even though the mixture cost (2.1) is not geodesically convex, for GMM optimization geodesic convexity seems to play a crucial role, and it has a huge impact on convergence speed. This behavior is partially expected and analogous to EM, where a convex M-Step makes the overall method much more practical. Let us now use this intuition to elicit geodesic convexity.

## 2.2 Problem reformulation

We begin with parameter estimation for a single Gaussian: although this has a closed-form solution (which ultimately benefits EM), it requires more subtle handling when using manifold optimization. Consider the following maximum likelihood parameter estimation for a single Gaussian:

$$\max_{\boldsymbol{\mu}, \boldsymbol{\Sigma} \succ 0} \mathcal{L}(\boldsymbol{\mu}, \boldsymbol{\Sigma}) := \sum\nolimits_{i=1}^{n} \log p_{\mathcal{N}}(\boldsymbol{x}_i; \boldsymbol{\mu}, \boldsymbol{\Sigma}). \tag{2.2}$$

Although (2.2) is a Euclidean convex problem, it is *not* geodesically convex on its domain $\mathbb{R}^d \times \mathbb{P}^d$, which makes it geometrically handicapped when applying manifold optimization. To overcome this problem, we invoke a simple reparametrization[6] that has far-reaching impact. More precisely, we augment the sample vectors $\boldsymbol{x}_i$ to instead consider $\boldsymbol{y}_i^T = [\boldsymbol{x}_i^T \ 1]$. Therewith, (2.2) turns into

$$\max_{\boldsymbol{S} \succ 0} \widehat{\mathcal{L}}(\boldsymbol{S}) := \sum\nolimits_{i=1}^{n} \log q_{\mathcal{N}}(\boldsymbol{y}_i; \boldsymbol{S}), \tag{2.3}$$

where $q_{\mathcal{N}}(\boldsymbol{y}_i; \boldsymbol{S}) := \sqrt{2\pi} \exp(\frac{1}{2}) p_{\mathcal{N}}(\boldsymbol{y}_i; \boldsymbol{0}, \boldsymbol{S})$. Proposition 1 states the key property of (2.3).

**Proposition 1.** *The map* $\phi(\boldsymbol{S}) \equiv -\widehat{\mathcal{L}}(\boldsymbol{S})$, *where* $\widehat{\mathcal{L}}(\boldsymbol{S})$ *is as in* (2.3), *is geodesically convex.*

We omit the proof due to space limits; see [13] for details. Alternatively, see [28] for more general results on geodesic convexity.

Theorem 2.1 shows that the solution to (2.3) yields the solution to the original problem (2.2) too.

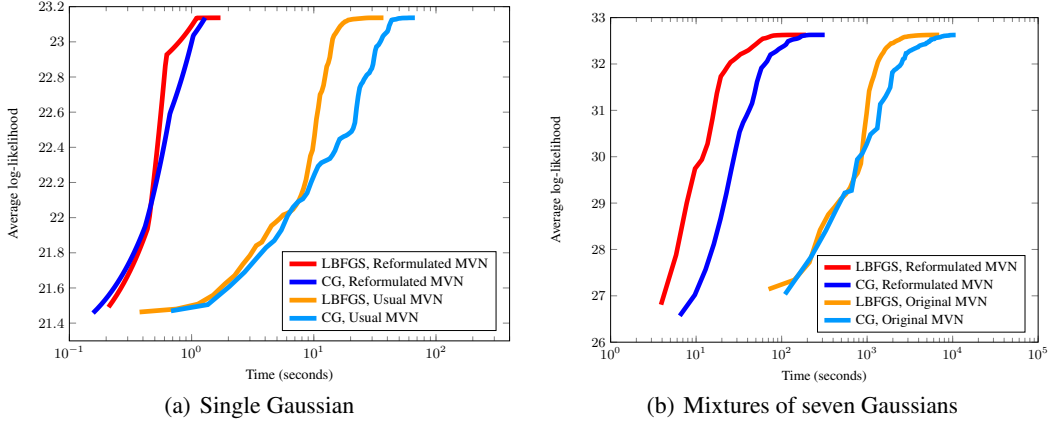

(a) Single Gaussian
(b) Mixtures of seven Gaussians

Figure 1: The effect of reformulation in convergence speed of manifold CG and manifold LBFGS methods ($d = 35$); note that the x-axis (time) is on a logarithmic scale.

**Theorem 2.1.** *If $\boldsymbol{\mu}^*, \boldsymbol{\Sigma}^*$ maximize* (2.2), *and if $\boldsymbol{S}^*$ maximizes* (2.3), *then $\widehat{\mathcal{L}}(\boldsymbol{S}^*) = \mathcal{L}(\boldsymbol{\mu}^*, \boldsymbol{\Sigma}^*)$ for*

$$\boldsymbol{S}^* = \begin{pmatrix} \boldsymbol{\Sigma}^* + \boldsymbol{\mu}^* \boldsymbol{\mu}^{*T} & \boldsymbol{\mu}^* \\ \boldsymbol{\mu}^{*T} & 1 \end{pmatrix}.$$

*Proof.* We express $\boldsymbol{S}$ by new variables $\boldsymbol{U}, \boldsymbol{t}$ and $s$ by writing $\boldsymbol{S} = \begin{pmatrix} \boldsymbol{U} + s\boldsymbol{t}\boldsymbol{t}^T & s\boldsymbol{t} \\ s\boldsymbol{t}^T & s \end{pmatrix}$. The objective function $\widehat{\mathcal{L}}(\boldsymbol{S})$ in terms of the new parameters becomes

$$\widehat{\mathcal{L}}(\boldsymbol{U}, \boldsymbol{t}, s) = \frac{n}{2} - \frac{d}{2}\log(2\pi) - \frac{n}{2}\log s - \frac{n}{2}\log\det(\boldsymbol{U}) - \sum_{i=1}^{n} \frac{1}{2}(\boldsymbol{x}_i - \boldsymbol{t})^T \boldsymbol{U}^{-1}(\boldsymbol{x}_i - \boldsymbol{t}) - \frac{n}{2s}.$$

Optimizing $\widehat{\mathcal{L}}$ over $s > 0$ we see that $s^* = 1$ must hold. Hence, the objective reduces to a $d$-dimensional Gaussian log-likelihood, for which clearly $\boldsymbol{U}^* = \boldsymbol{\Sigma}^*$ and $\boldsymbol{t}^* = \boldsymbol{\mu}^*$. □

Theorem 2.1 shows that reformulation (2.3) is "faithful," as it leaves the optimum unchanged. Theorem 2.2 proves a local version of this result for GMMs.

**Theorem 2.2.** *A local maximum of the reparameterized GMM log-likelihood*

$$\widehat{\mathcal{L}}(\{\boldsymbol{S}_j\}_{j=1}^K) := \sum_{i=1}^{n} \log\Big(\sum_{j=1}^{K} \alpha_j q_{\mathcal{N}}(\boldsymbol{y}_i; \boldsymbol{S}_j)\Big)$$

*is a local maximum of the original log-likelihood*

$$\mathcal{L}(\{\boldsymbol{\mu}_j, \boldsymbol{\Sigma}_j\}_{j=1}^K) := \sum_{i=1}^{n} \log\Big(\sum_{j=1}^{K} \alpha_j p_{\mathcal{N}}(\boldsymbol{x}_i | \boldsymbol{\mu}_j, \boldsymbol{\Sigma}_j)\Big).$$

The proof can be found in [13].

Theorem 2.2 shows that we can replace problem (2.1) by one whose local maxima agree with those of (2.1), and whose individual components are geodesically convex. Figure 1 shows the true import of our reformulation: the dramatic impact on the empirical performance of Riemmanian Conjugate-Gradient (CG) and Riemannian LBFGS for GMMs is unmistakable.

The final technical piece is to replace the simplex constraint $\boldsymbol{\alpha} \in \Delta_K$ to make the problem unconstrained. We do this via a commonly used change of variables [14]: $\eta_k = \log\big(\frac{\alpha_k}{\alpha_K}\big)$ for $k = 1, \dots, K-1$. Assuming $\eta_K = 0$ is a constant, the final GMM optimization problem is:

$$\max_{\{\boldsymbol{S}_j \succ 0\}_{j=1}^K, \{\eta_j\}_{j=1}^{K-1}} \widehat{\mathcal{L}}(\{\boldsymbol{S}_j\}_{j=1}^K, \{\eta_j\}_{j=1}^{K-1}) := \sum_{i=1}^{n} \log\Big(\sum_{j=1}^{K} \frac{\exp(\eta_j)}{\sum_{k=1}^{K} \exp(\eta_k)} q_{\mathcal{N}}(\boldsymbol{y}_i; \boldsymbol{S}_j)\Big) \quad (2.4)$$

We view (2.4) as a manifold optimization problem; specifically, it is an optimization problem on the product manifold $\big(\prod_{j=1}^{K} \mathbb{P}^d\big) \times \mathbb{R}^{K-1}$. Let us see how to solve it.

# 3 Manifold Optimization

In unconstrained Euclidean optimization, typically one iteratively (i) finds a descent direction; and (ii) performs a line-search to obtain sufficient decrease and ensure convergence. On a Riemannian manifold, the descent direction is computed on the tangent space (this space varies (smoothly) as one moves along the manifold). At a point $X$, the tangent space $T_X$ is the approximating vector space (see Fig. 2). Given a descent direction $\xi_X \in T_X$, line-search is performed along a smooth curve on the manifold (red curve in Fig. 2). The derivative of this curve at $X$ equals the descent direction $\xi_X$. We refer the reader to [1, 29] for an in depth introduction to manifold optimization.

Successful large-scale Euclidean methods such as conjugate-gradient and LBFGS combine gradients at the current point with gradients and descent directions from previous points to obtain a new descent direction. To adapt such algorithms to manifolds, in addition to defining gradients on manifolds, we also need to define how to transport vectors in a tangent space at one point to vectors in a different tangent space at another point.

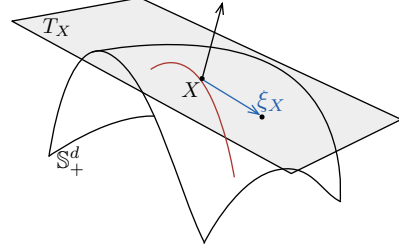

On Riemannian manifolds, the gradient is simply a direction on the tangent space, where the inner-product of the gradient with another direction in the tangent space gives the directional derivative of the function. Formally, if $g_X$ defines the inner product in the tangent space $T_X$, then

$$Df(X)\xi = g_X(\mathrm{grad} f(X), \xi), \quad \text{for } \xi \in T_X.$$

Figure 2: Visualization of line-search on a manifold: $X$ is a point on the manifold, $T_X$ is the tangent space at the point $X$, $\xi_X$ is a descent direction at $X$; the red curve is the curve along which line-search is performed.

Given a descent direction in the tangent space, the curve along which we perform line-search can be a geodesic. A map that takes the direction and a step length to obtain a corresponding point on the geodesic is called an *exponential map*. Riemannian manifolds are also equipped with a natural way of transporting vectors on geodesics, which is called parallel transport. Intuitively, a parallel transport is a differential map with zero derivative along the geodesics. Using the above ideas, Algorithm 1 sketches a generic manifold optimization algorithm.

---

Algorithm 1: Sketch of an optimization algorithm (CG, LBFGS) to minimize $f(X)$ on a manifold

---

**Given:** Riemannian manifold $\mathcal{M}$ with Riemannian metric $g$; parallel transport $\mathcal{T}$ on $\mathcal{M}$; exponential map $R$; initial value $X_0$; a smooth function $f$
**for** $k = 0, 1, \ldots$ **do**
    Obtain a descent direction based on stored information and $\mathrm{grad} f(X_k)$ using metric $g$ and transport $\mathcal{T}$
    Use line-search to find $\alpha$ such that it satisfies appropriate (descent) conditions
    Calculate the retraction / update $X_{k+1} = R_{X_k}(\alpha\xi_k)$
    Based on the memory and need of algorithm store $X_k$, $\mathrm{grad} f(X_k)$ and $\alpha\xi_k$
**end for**
**return** estimated minimum $X_k$

---

Note that Cartesian products of Riemannian manifolds are again Riemannian, with the exponential map, gradient and parallel transport defined as the Cartesian product of individual expressions; the inner product is defined as the sum of inner product of the components in their respective manifolds. Different variants of Riemannian LBFGS can be obtained depending where to perform the vector

| Definition | Expression for PSD matrices |
|---|---|
| Tangent space | Space of symmetric matrices |
| Metric between two tangent vectors $\xi, \eta$ at $\Sigma$ | $g_\Sigma(\xi, \eta) = \mathrm{tr}(\Sigma^{-1}\xi\Sigma^{-1}\eta)$ |
| Gradient at $\Sigma$ if Euclidean gradient is $\nabla f(\Sigma)$ | $\mathrm{grad} f(\Sigma) = \frac{1}{2}\Sigma(\nabla f(X) + \nabla f(X)^T)\Sigma$ |
| Exponential map at point $\Sigma$ in direction $\xi$ | $R_\Sigma(\xi) = \Sigma \exp(\Sigma^{-1}\xi)$ |
| Parallel transport of tangent vector $\xi$ from $\Sigma_1$ to $\Sigma_2$ | $\mathcal{T}_{\Sigma_1,\Sigma_2}(\xi) = E\xi E^T, \quad E = (\Sigma_2\Sigma_1^{-1})^{1/2}$ |

Table 1: Summary of key Riemannian objects for the PD matrix manifold.

transport. We found that the version developed in [28] gives the best performance, once we combine it with a line-search algorithm satisfying Wolfe conditions. We present the crucial details below.

### 3.1  Line-search algorithm satisfying Wolfe conditions

To ensure Riemannian LBFGS always produces a descent direction, it is necessary to ensure that the line-search algorithm satisfies Wolfe conditions [25]. These conditions are given by:

$$f(R_{X_k}(\alpha\xi_k)) \leq f(X_k) + c_1\alpha Df(X_k)\xi_k, \tag{3.1}$$

$$Df(X_{k+1})\xi_{k+1} \geq c_2 Df(X_k)\xi_k, \tag{3.2}$$

where $0 < c_1 < c_2 < 1$. Note that $\alpha Df(X_k)\xi_k = g_{X_k}(\text{grad} f(X_k), \alpha\xi_k)$, i.e., the derivative of $f(X_k)$ in the direction $\alpha\xi_k$ is the inner product of descent direction and gradient of the function. Practical line-search algorithms implement a stronger (Wolfe) version of (3.2) that enforces

$$|Df(X_{k+1})\xi_{k+1}| \leq c_2 Df(X_k)\xi_k.$$

Similar to the Euclidean case, our line-search algorithm is also divided into two phases: bracketing and zooming [23]. During bracketing, we compute an interval such that a point satisfying Wolfe conditions can be found in this interval. In the zooming phase, we obtain such a point in the determined interval. The one-dimensional function and its gradient used by the line-search are

$$\phi(\alpha) = f(R_{X_k}(\alpha\xi_k)), \qquad \phi'(\alpha) = \alpha Df(X_k)\xi_k.$$

The algorithm is essentially the same as the line-search in the Euclidean space; the reader can also see its manifold incarnation in [13]. Theory behind how this algorithm is guaranteed to find a step-length satisfying (strong) Wolfe conditions can be found in [23].

A good choice of initial step-length $\alpha_1$ can greatly speed up the line-search. We propose the following choice that turns out to be quite effective in our experiments:

$$\alpha_1 = 2\frac{f(X_k) - f(X_{k-1})}{Df(X_k)\xi_k}. \tag{3.3}$$

Equation (3.3) is obtained by finding $\alpha^*$ that minimizes a quadratic approximation of the function along the geodesic through the previous point (based on $f(X_{k-1})$, $f(X_k)$ and $Df(X_{k-1})\xi_{k-1}$):

$$\alpha^* = 2\frac{f(X_k) - f(X_{k-1})}{Df(X_{k-1})\xi_{k-1}}. \tag{3.4}$$

Then assuming that first-order change will be the same as in the previous step, we write

$$\alpha^* Df(X_{k-1})\xi_{k-1} \approx \alpha_1 Df(X_k)\xi_k. \tag{3.5}$$

Combining (3.4) and (3.5), we obtain our estimate $\alpha_1$ expressed in (3.3). Nocedal and Wright [23] suggest using either $\alpha^*$ of (3.4) for the initial step-length $\alpha_1$, or using (3.5) where $\alpha^*$ is set to be the step-length obtained in the line-search in the previous point. We observed that if one instead uses (3.3) instead, one obtains substantially better performance than the other two approaches.

## 4  Experimental Results

We have performed numerous experiments to examine effectiveness of our method. Below we report performance comparisons on both real and simulated data. In all experiments, we initialize the mixture parameters for all methods using k-means++ [2]. All methods also use the same termination criteria: they stop either when the difference of *average log-likelihood* (i.e., $\frac{1}{n}$log-likelihood) between consecutive iterations falls below $10^{-6}$, or when the number of iterations exceeds 1500. More extensive empirical results can be found in the longer version of this paper [13].

**Simulated Data**

EM's performance is well-known to depend on the degree of separation of the mixture components [18, 34]. To assess the impact of this separation on our methods, we generate data as proposed in [8, 32]. The distributions are chosen so their means satisfy the following inequality:

$$\forall_{i\neq j} : \|\boldsymbol{m}_i - \boldsymbol{m}_j\| \geq c\max_{i,j}\{\text{tr}(\boldsymbol{\Sigma}_i), \text{tr}(\boldsymbol{\Sigma}_j)\},$$

where $c$ models the degree of separation. Since mixtures with high eccentricity (i.e., the ratio of the largest eigenvalue of the covariance matrix to its smallest eigenvalue) have smaller overlap, in

| | | EM Original | | LBFGS Reformulated | | CG Reformulated | | CG Original | |
|---|---|---|---|---|---|---|---|---|---|
| | | Time (s) | ALL | Time (s) | ALL | Time (s) | ALL | Time (s) | ALL |
| $c = 0.2$ | $K = 2$ | $1.1 \pm 0.4$ | -10.7 | $5.6 \pm 2.7$ | -10.7 | $3.7 \pm 1.5$ | -10.8 | $23.8 \pm 23.7$ | -10.7 |
| | $K = 5$ | $30.0 \pm 45.5$ | -12.7 | $49.2 \pm 35.0$ | -12.7 | $47.8 \pm 40.4$ | -12.7 | $206.0 \pm 94.2$ | -12.8 |
| $c = 1$ | $K = 2$ | $0.5 \pm 0.2$ | -10.4 | $3.1 \pm 0.8$ | -10.4 | $2.6 \pm 0.6$ | -10.4 | $25.6 \pm 13.6$ | -10.4 |
| | $K = 5$ | $104.1 \pm 113.8$ | -13.4 | $79.9 \pm 62.8$ | -13.3 | $45.8 \pm 30.4$ | -13.3 | $144.3 \pm 48.1$ | -13.3 |
| $c = 5$ | $K = 2$ | $0.2 \pm 0.2$ | -11.0 | $3.4 \pm 1.4$ | -11.0 | $2.8 \pm 1.2$ | -11.0 | $43.2 \pm 38.8$ | -11.0 |
| | $K = 5$ | $38.8 \pm 65.8$ | -12.8 | $41.0 \pm 45.7$ | -12.8 | $29.2 \pm 36.3$ | -12.8 | $197.6 \pm 118.2$ | -12.8 |

Table 2: Speed and average log-likelihood (ALL) comparisons for $d = 20$, $e = 10$ (each row reports values averaged over 20 runs over *different* datasets, so the ALL values are not comparable to each other).

| | | EM Original | | LBFGS Reformulated | | CG Reformulated | | CG Original | |
|---|---|---|---|---|---|---|---|---|---|
| | | Time (s) | ALL | Time (s) | ALL | Time (s) | ALL | Time (s) | ALL |
| $c = 0.2$ | $K = 2$ | $65.7 \pm 33.1$ | 17.6 | $39.4 \pm 19.3$ | 17.6 | $46.4 \pm 29.9$ | 17.6 | $64.0 \pm 50.4$ | 17.6 |
| | $K = 5$ | $365.6 \pm 138.8$ | 17.5 | $160.9 \pm 65.9$ | 17.5 | $207.6 \pm 46.9$ | 17.5 | $279.8 \pm 169.3$ | 17.5 |
| $c = 1$ | $K = 2$ | $6.0 \pm 7.1$ | 17.0 | $12.9 \pm 13.0$ | 17.0 | $15.7 \pm 17.5$ | 17.0 | $42.5 \pm 21.9$ | 17.0 |
| | $K = 5$ | $40.5 \pm 61.1$ | 16.2 | $51.6 \pm 39.5$ | 16.2 | $63.7 \pm 45.8$ | 16.2 | $203.1 \pm 96.3$ | 16.2 |
| $c = 5$ | $K = 2$ | $0.2 \pm 0.1$ | 17.1 | $3.0 \pm 0.5$ | 17.1 | $2.8 \pm 0.7$ | 17.1 | $19.6 \pm 8.2$ | 17.1 |
| | $K = 5$ | $17.5 \pm 45.6$ | 16.1 | $20.6 \pm 22.5$ | 16.1 | $20.3 \pm 24.1$ | 16.1 | $93.9 \pm 42.4$ | 16.1 |

Table 3: Speed and ALL comparisons for $d = 20$, $e = 1$.

| | | CG Cholesky Original | | | | CG Cholesky Reformulated | | | |
|---|---|---|---|---|---|---|---|---|---|
| | | $e = 1$ | | $e = 10$ | | $e = 1$ | | $e = 10$ | |
| | | Time (s) | ALL | Time (s) | ALL | Time (s) | ALL | Time (s) | ALL |
| $c = 0.2$ | $K = 2$ | $101.5 \pm 34.1$ | 17.6 | $113.9 \pm 48.1$ | -10.7 | $36.7 \pm 9.8$ | 17.6 | $23.5 \pm 11.9$ | -10.7 |
| | $K = 5$ | $627.1 \pm 247.3$ | 17.5 | $521.9 \pm 186.9$ | -12.7 | $156.7 \pm 81.1$ | 17.5 | $106.7 \pm 39.7$ | -12.6 |
| $c = 1$ | $K = 2$ | $135.2 \pm 65.4$ | 16.9 | $110.9 \pm 51.8$ | -10.4 | $38.0 \pm 14.5$ | 16.9 | $49.0 \pm 17.8$ | -10.4 |
| | $K = 5$ | $1016.9 \pm 299.8$ | 16.2 | $358.0 \pm 155.5$ | -13.3 | $266.7 \pm 140.5$ | 16.2 | $279.8 \pm 111.0$ | -13.4 |
| $c = 5$ | $K = 2$ | $55.2 \pm 27.9$ | 17.1 | $86.7 \pm 47.2$ | -11.0 | $60.2 \pm 20.8$ | 17.1 | $177.6 \pm 147.6$ | -11.0 |
| | $K = 5$ | $371.7 \pm 281.4$ | 16.1 | $337.7 \pm 178.4$ | -12.8 | $270.2 \pm 106.5$ | 16.1 | $562.1 \pm 242.7$ | -12.8 |

Table 4: Speed and ALL for applying CG on Cholesky-factorized problems with $d = 20$.

addition to high eccentricity $e = 10$, we also test the (spherical) case where $e = 1$. We test three levels of separation $c = 0.2$ (low), $c = 1$ (medium), and $c = 5$ (high). We test two different numbers of mixture components $K = 2$ and $K = 5$; we consider experiments with larger values of $K$ in our experiments on real data. For $e = 10$, the results for data with dimensionality $d = 20$ are given in Table 2. The results are obtained after running with 20 different random choices of parameters for each configuration. It is apparent that the performance of EM and Riemannian optimization with our reformulation is very similar. The variance of computation time shown by Riemmanian optimization is, however, notably smaller. Manifold optimization on the non-reformulated problem (last column) performs the worst.

In another set of simulated data experiments, we apply different algorithms to spherical data ($e = 1$); the results are shown in Table 3. The interesting instance here is the case of low separation $c = 0.2$, where the condition number of the Hessian becomes large. As predicted by theory, the EM converges very slowly in such a case; Table 3 confirms this claim. It is known that in this case, the performance of powerful optimization approaches like CG and LBFGS also degrades [23]. But both CG and LBFGS suffer less than EM, while LBFGS performs noticeably better than CG.

Cholesky decomposition is a commonly suggested idea for dealing with PD constraint. So, we also compare against unconstrained optimization (using Euclidean CG), where the inverse covariance matrices are Cholesky factorized. The results for the same data as in Tables 2 and 3 are reported in Table 4. Although the Cholesky-factorized problem proves to be much inferior to both EM and the manifold methods, our reformulation seems to also help it in several problem instances.

**Real Data**

We now present performance evaluation on a natural image dataset, where mixtures of Gaussians were reported to be a good fit to the data [35]. We extracted 200,000 image patches of size $6 \times 6$ from images and subtracted the DC component, leaving us with 35-dimensional vectors. Performance of different algorithms are reported in Table 5. Similar to the simulated results, performance of EM and

|  | EM Algorithm | | LBFGS Reformulated | | CG Reformulated | | CG Original | | CG Cholesky Reformulated | |
|---|---|---|---|---|---|---|---|---|---|---|
|  | Time (s) | ALL | Time (s) | ALL | Time (s) | ALL | Time (s) | ALL | Time (s) | ALL |
| $K = 2$ | 16.61 | 29.28 | 14.23 | 29.28 | 17.52 | 29.28 | 947.35 | 29.28 | 476.77 | 29.28 |
| $K = 3$ | 90.54 | 30.95 | 38.29 | 30.95 | 54.37 | 30.95 | 3051.89 | 30.95 | 1046.61 | 30.95 |
| $K = 4$ | 165.77 | 31.65 | 106.53 | 31.65 | 153.94 | 31.65 | 6380.01 | 31.64 | 2673.21 | 31.65 |
| $K = 5$ | 202.36 | 32.07 | 117.14 | 32.07 | 140.21 | 32.07 | 5262.27 | 32.07 | 3865.30 | 32.07 |
| $K = 6$ | 228.80 | 32.36 | 245.74 | 32.35 | 281.32 | 32.35 | 10566.76 | 32.33 | 4771.36 | 32.35 |
| $K = 7$ | 365.28 | 32.63 | 192.44 | 32.63 | 318.95 | 32.63 | 10844.52 | 32.63 | 6819.42 | 32.63 |
| $K = 8$ | 596.01 | 32.81 | 332.85 | 32.81 | 536.94 | 32.81 | 14282.80 | 32.58 | 9306.33 | 32.81 |
| $K = 9$ | 900.88 | 32.94 | 657.24 | 32.94 | 1449.52 | 32.95 | 15774.88 | 32.77 | 9383.98 | 32.94 |
| $K = 10$ | 2159.47 | 33.05 | 658.34 | 33.06 | 1048.00 | 33.06 | 17711.87 | 33.03 | 7463.72 | 33.05 |

Table 5: Speed and ALL comparisons for natural image data $d = 35$.

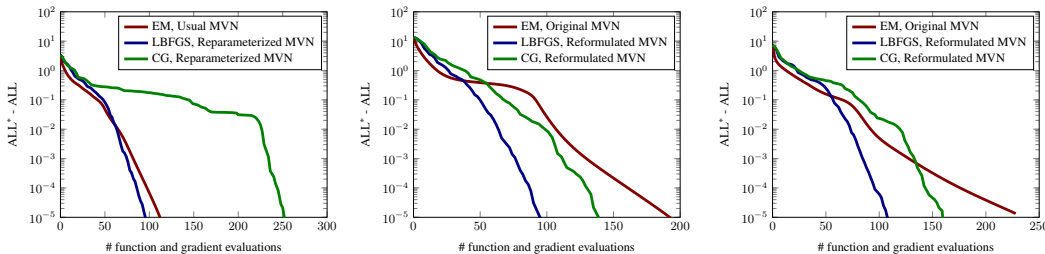

Figure 3: Best ALL minus current ALL values with number of function and gradient evaluations. Left: 'magic telescope' ($K = 5, d = 10$). Middle: 'year predict' ($K = 6, d = 90$). Right: natural images ($K = 8, d = 35$).

manifold CG on the reformulated parameter space is similar. Manifold LBFGS converges notably faster (except for $K = 6$) than both EM and CG. Without our reformulation, performance of the manifold methods degrades substantially. Note that for $K = 8$ and $K = 9$, CG without reformulation stops prematurely because it hits the bound of a maximum 1500 iterations, and therefore its ALL is smaller than the other two methods. The table also shows results of the Cholesky-factorized (and reformulated) problem. It is more than 10 times slower than manifold optimization. Optimizing the Cholesky-factorized (non-reformulated) problem is the slowest (not shown) and it always reaches the maximum number of iterations before finding the local minimum.

Fig. 3 depicts the typical behavior of our manifold optimization methods versus EM. The X-axis is the number of log-likelihood and gradient evaluations (or the number of E- and M-steps in EM). Fig. 3(a) and Fig. 3(b) are the results of fitting GMMs to the 'magic telescope' and 'year prediction' datasets[7]. Fig. 3(c) is the result for the natural image data of Table 5. Apparently in the initial few iterations EM is faster, but manifold optimization methods match EM in a few iterations. This is remarkable, given that manifold optimization methods need to perform line-search.

## 5 Conclusions and future work

We introduced Riemannian manifold optimization as an alternative to EM for fitting Gaussian mixture models. We demonstrated that for making manifold optimization succeed, to either match or outperform EM, it is necessary to represent the parameters in a different space and reformulate the cost function accordingly. Extensive experimentation with both experimental and real datasets yielded quite encouraging results, suggesting that manifold optimization could have the potential to open new algorithmic avenues for mixture modeling.

Several strands of practical importance are immediate (and are a part of our ongoing work): (i) extension to large-scale GMMs through stochastic optimization [5]; (ii) use of richer classes of priors with GMMs than the usual inverse Wishart priors (which are typically also used as they make the M-step convenient), which is actually just one instance of a geodesically convex prior that our methods can handle; (iii) incorporation of penalties for avoiding tiny clusters, an idea that fits easily in our framework but not so easily in the EM framework. Finally, beyond GMMs, extension to other mixture models will be fruitful.

## Footnotes

[1]Remarkably, using Cholesky with the reformulation in §2.2 does not add spurious local minima to GMMs.

[2]Equivalently, on the interior of the constraint set, as is done by interior point methods (their nonconvex versions); though these turn out to be slow too as they are second order methods.

[3]That is, convex along geodesic curves on the PD manifold.

[4]Manifold optimization should not be confused with "manifold learning" a separate problem altogether.

[5]Though under very strong assumptions, it has polynomial smoothed complexity [11].

[6]This reparametrization in itself is probably folklore; its role in GMM optimization is what is crucial here.

[7] Available at UCI machine learning dataset repository via https://archive.ics.uci.edu/ml/datasets

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
