[Reviews · NeurIPS 2015]

Submitted by Assigned_Reviewer_1

This paper revisits fitting mixtures of Gaussians with gradient methods. A combination of manifold optimization, with a particular linesearch procedure, and a "reparameterized" objective, empirically beats batch EM in the experiments reported.

While I'm not an expert in Riemannian manifolds, the work appears plausible. The work is a reported at a level that's possible to follow, if taking results from cited references on trust.

Plain mixtures of Gaussians may not seem particularly exciting in this world of Bayesian nonparametrics and deep neural networks. However, GMMs are quietly plodding along as important work-horses in applications: they're useful and surprisingly powerful. People often then want to extend GMMs in various ways, e.g., by adding constraints or tying covariances together. But then EM usually doesn't apply. Gradient-based procedures would generalize fine, so ones that work well on the base model could have significant impact on statistical applications.

I would previously have used an unconstrained parameterization based on the Cholesky. While the paper gives reasons to disfavour this approach, it's a shame it's not included in the empirical comparison. Are spurious stationary points an issue in practice? Is the manifold optimizer actually better? Is the main gain here from the "representation" in (2.3); would that also help the Cholesky approach? The paper would be stronger if it answered these questions.

Minor comments:

Line 081: "open a new" -> "open new"

Line 138: rely *on* geodesics

The timings in Table 4 would be probably be better as a graph. A graph would give an idea of the comparison more quickly. Precise times are irrelevant, being implementation and machine dependent.

I suggest replacing lines 147-149 "At any point... specifically [4]" with: "Geodesics on the manifold from $\Sigma_1$ to $\Sigma_2$ are given by [4]:". Riemannian metrics are not explained in the paper, and don't need to be understood or mentioned to take on trust the geodesic result. (The lack of context, and the nearby definition of `$d$', encourges a misreading of $d$ as a variable, making the trace expression evaluate to `$d^3$' (!).)

I suggest adding the zero mean ("0,") explicitly to the normal density in (2.3) and the line below it. On a quick reading, augmenting the data with a constant in the line above (2.3) seems like a terrible idea, a Gaussian could fit these new data with infinite density. The zero-mean constraint is what stops that from happening. Also, the compact Gaussian notation is only defined explicitly including a mean term above (2.1).

I wouldn't call the transformed problem (2.3) a "reparameterization". A different model is being fitted, one that would generate data from a different distribution than the original.

I suggest numbering all equations so other people can refer to them, not just in reviews, but in reading groups and citations.
Summary: Novel gradient-based procedure for fitting GMMs, which appears to beat EM. While I'm slightly unclear on some claims, I'm interested in this paper as potentially really useful in applications.

I have increased my score in response to the rebuttal. The authors promise further clarification and control experiments to compare this work to the obvious Cholesky approach. I think this will be an interesting paper.

Submitted by Assigned_Reviewer_2

summary: manifold optimization of GMMs

Quality: high (i think, to be determined)

Clarity: high

Originality: seems so to me (though i'm unfamiliar with the manifold optimization literature)

Significance: good good.

i like this paper. my comments are as follows:

1) i don't like tables, and i don't like only seeing means & standard errors, especially for hard optimization problems.

i also don't like seeing only performance times, because i can't tell how well the algorithms performed in terms of the objective.

2) i'd like the benchmarks from UCI to be in the main text. i'd also like to see comparisons with other implementations, eg, if you are using matlab, perhaps the gmmdist set of functions or mixmod, or in R, perhaps mclust.

3) in general, saying "EM algorithm" as your benchmark it is unclear what implementation you are using, which matters a lot.

for example, setting the convergence criteria differently can make the run time quite different, but the estimates quite similar.

therefore, i'd like to see the following:

(a) two-dimensional contour plots (or some such), showing both time and performance for each of the numerical examples.

(b) same thing for a set of benchmarks.

(c) some plots showing likelihood (or the surrogate objective) and time as a function of iteration.

one big problem with EM is that often, the loss gets nearly flat, so the last bunch of iterations are useless, and the termination criteria could be set differently to obtain vastly superior performance.

i don't think it matters much what the results are, as long as they are qualitatively consistent with the text. but currently, the manuscript does not provide any evidence that the algorithms actually work.

Summary: developed a modern algorithm for a classic problem, with promising results, however, no accuracy results are provided,

nor details of the benchmark algorithms, nor performance vs iteration, so more details are required for proper assessment.

Submitted by Assigned_Reviewer_3

Learning Gaussian Mixture Models is a very well-studied problem in statistics and ML. Expectation Maximization (EM) algorithm from 30 years ago still remains the most popular algorithm. The current paper proposes to use manifold optimization (a reasonably well-established subarea of optimization) for this problem. Very briefly, think of the optimization problem where the variables are the parameters of the mixture model (i.e., the mean, covariance and weight for each component, where the number of components is fixed), and given these one can write down the expression for the (log-) likelihood of the given data. The most important constraint on the variables---and the main source of difficulty---is that the covariance matrices are positive semi-definite. One seeks to find parameters that maximize this. As the paper observes, direct use of standard manifold optimization is not effective as the algorithms tend to be too slow. But, as the paper shows, a simple re-parametrization of the optimization program plus judicious application of standard manifold optimization algorithms with some tweaking improves the performance greatly: In the reported experiments, their algorithm (the paper has several algorithms; I will focus on the best) converges in time that's comparable and often better than EM, and the final value of the objective functions is generally the same for the two algorithms. (Note that these methods may converge to local maxima, and the question of how far the parameters thus estimated can be from the true parameters is not discussed here.) It of course remains to be seen how widely this holds. But given the importance of GMM, these results are interesting.

Minor: Line 129: "Problem (2.1) in general can require exponential time." This should be phrased more carefully: "Problem (2.1) in general can require number of samples that's exponential in K."
Summary: I think the paper can be accepted on the strength of experimental results.

Author Feedback
Author rebuttal: We thank the reviewers for their feedback, suggestions and encouragement. Before we address the comments individually, we'd like to emphasize our message:

Matching or exceeding EM on GMMs using manifold optimization methods is valuable, especially because the flexibility of these methods opens new algorithmic avenues.

Reviewer 1

- Reparametrization
We agree that reparametrization may not be fully correct. As shown in Theorem 2.2, the new objective function is a reparametrization of the original at its stationary points. To avoid confusion we can call it model-transformation.

- Cholesky
We apologize that we extrapolated based on comments in [29] to claim that Cholesky factorization adds spurious stationary points. In fact, we realized that our model-transformation shows that Cholesky does not add any local minima! (this observation may be new). Curiously, our transformation also accelerates convergence of Cholesky formulation.

But even the transformd Cholesky problem is 2X-10X slower than our manifold approach. During the rebuttal period, we could only finish simulations of Cholesky approach for Table 1. We will update our paper with more complete results.

Supplement to Table 1 [Time(s)]
| CG Cholesky | LBFGS Cholesky|
---------------------------------
c=0.2 K=2 | 23.5+-11.9 | 14.7+-6.4 |
...... K=5 | 106.7+-39.7 | 68.6+-41.4 |

c=1 K=2 | 49.0+-17.8 | 31.8+-16.3 |
... K=5 | 279.8+-111 | 272.1+-137 |

c=5 K=2 | 177.6+-147.6 | 117.7+-58 |
......K=5 | 562.1+-242.7 | 606.7+-344|

Thanks for careful reading, catching typos, and other suggestions. We will apply all suggestions in the final version.

Reviewer 2

- 1st comment
Since the reviewer suggests contour plots for the benchmark data, perhaps the reviewer thinks we are using different initializations and showing averaged values on benchmark data? For this data, we ran the algorithms only once and report the objective and the performance time. The objective ALL (average log-likelihood) is the log-likelihood of the data divided by the number of samples. Since the ALL values can be arbitrary, contour plots may be quite noisy and not so informative.

Similar to EM, manifold optimization converges fast in the first few iterations. We will add time/likelihood/iteration plots as per the reviewer's suggestion in (c). One concrete such result is the following (corr. to d=35, and K=5; iteration means one step of EM, or one LL and gradient evaluation for LBFGS):

Iter | LBFGS | EM |
2 | 25.78 | 30.18|
8 | 31.40 | 31.51|
16 | 32.05 | 31.80|
32 | 32.07 | 31.98|
64 | - | 32.07|

It is true that EM gets slow close to singular solutions; but the same is true for other optimization methods. We sought a fair stopping criterion and used a tolerance of 1e-6, which seems to be a standard value in many implementations, including Matlab. For a looser criterion, in flatter regions LBFGS may stop early and yield smaller LL, but EM suffers more. For tighter criteria, unsurprisingly EM suffers much more.

- 2nd Comment
Computation costs for EM are just the E-step and M-step, while the main costs for manifold optimization (esp. for high-D) are computing log-likelihood and gradients. The E-step costs similar to computing the log-likelihood; the M-step costs about the same as the gradient, so mismatch due to implementation is reasonably small.

We developed a Matlab toolbox for mixture models, and used it in our simulations; we will add a link to this toolbox. Upon the reviewer's suggestion, we also tried 'fitgmdist' from Matlab; the timing results are very similar. The results reported in our paper are very conservative: indeed, the implementation of the manifold optimization methods can be substantially improved, which may not be the case for EM.

We assure that we will try our best to squeeze UCI results into the main text.

- 3rd Comment
As noted above, we believe we used a fair convergence criterion; we can add results with looser and tighter criteria to offer a more complete picture.

Reviewer 3

Indeed, the methods may converge to local maxima. Avoiding local maxima is an independent research problem and many empirically effective approaches for tackling local maxima (eg. split-and-merge algorithm) exist; these currently focus on EM, but actually apply to our algorithms too.

Reviewer 4

We believe our paper offers a new set of algorithmic tools for GMMs. Our method is grounded in optimization, so extensions to stochastic, parallel, and other regimes are natural sequels.

Detailed comparisons with Z&W are orthogonal to our paper: we are proposing a new way to estimate GMM parameters, while Z&W use GMMs on natural images. We use image data as a realistic example on which people use GMMs. We have benchmarked our algorithm on several other data (see also appendix).